# Patch-Transformer Network: A Wearable-Sensor-Based Fall Detection Method

**DOI:** 10.3390/s23146360

**Published:** 2023-07-13

**Authors:** Shaobing Wang, Jiang Wu

**Affiliations:** School of Information Science and Engineering, Zhejiang Sci-Tech University, Hangzhou 310018, China; 202130504166@mail.zstu.edu.cn

**Keywords:** fall detection, deep learning, feature extraction, CNN, Transformer encoder

## Abstract

Falls can easily cause major harm to the health of the elderly, and timely detection can avoid further injuries. To detect the occurrence of falls in time, we propose a new method called Patch-Transformer Network (PTN) wearable-sensor-based fall detection algorithm. The neural network includes a convolution layer, a Transformer encoding layer, and a linear classification layer. The convolution layer is used to extract local features and project them into feature matrices. After adding positional coding information, the global features of falls are learned through the multi-head self-attention mechanism in the Transformer encoding layer. Global average pooling (GAP) is used to strengthen the correlation between features and categories. The final classification results are provided by the linear layer. The accuracy of the model obtained on the public available datasets SisFall and UnMib SHAR is 99.86% and 99.14%, respectively. The network model has fewer parameters and lower complexity, with detection times of 0.004 s and 0.001 s on the two datasets. Therefore, our proposed method can timely and accurately detect the occurrence of falls, which is important for protecting the lives of the elderly.

## 1. Introduction

According to the World Health Organization, approximately 28–35% of older people fall each year, increasing to 32–42% for those over 70 years of age [1]. Falls can cause multi-organ damage, such as brain, soft tissue, fracture, and dislocated joints, and are a major cause of disability or death in elderly [2]. Falls can also damage the confidence of older people, causing them to fear falling again [3]. Timely detection of falls and medical intervention are important to reduce physical damage and economic loss to the elderly.

In recent years, fall prediction and fall detection have been the main areas of research regarding falls [4]. Falls are caused by a variety of factors, physiological and environmental, with physiological factors referring to a person’s age, history of falls (e.g., plantar phobia), mobility disorders, sleep disorders, and neurological disorders. Environmental factors include smooth surfaces, dim light, etc. [5]. Fall prediction requires consideration of all influencing factors and does not prevent falls from occurring. Fall prediction has a high rate of false alarms and fall risk assessment can only be used as a reference for healthcare professionals [6]. Therefore, fall detection is the main solution to deal with the occurrence of fall events. Existing fall detection systems are similar in structure and their main purpose is to identify fall events and activities of daily living (ADLs) [7]. We propose a new deep learning fall detection method based on wearable sensors. Our approach is inspired by Transformer [8] and combines the characteristics of fall detection data samples to accurately distinguish between fall behavior and ADLs, and also has low temporal and spatial complexity, meeting the timeliness requirements of fall detection.

In this paper, we discuss existing methods regarding fall detection in Section 2, with a focus on wearable-sensor-based devices. In Section 3, we present the data used in the experiments, the data preprocessing steps, and details of the proposed PTN approach. In Section 4, the experimental setup and preparation are listed. The detection results and performance of this method are provided and compared with other existing fall detection methods. We briefly analyze feature extraction and model complexity in Section 5. In the end, Section 6 concludes the work.

## 2. Related Work

In recent years, research on fall detection has included vision-sensor-based [9,10], environmental-sensor-based [11,12], and wearable-device-based [13,14] methods. Among them, environmental sensors and vision sensors are both non-wearable sensors.

### 2.1. Non-Wearable Sensors

#### 2.1.1. Vision-Based Sensors

A vision-sensor-based fall detection system uses a camera as a data source to determine whether a fall has occurred. A camera [15] extracts information such as body posture shape, movement patterns, and movement of key points of the human skeleton [16]. Caroline used a combination of history and changing in human posture to detect falls and ADLs in video sequences [17]. Agrawal [18] used a modified gaussian mixture model (GMM) for background subtraction to find foreground objects and calculated the distance from the top of the rectangle covering the human body to the center of the middle to determine a fall. The user does not need to wear additional equipment. RGB-camera-based solutions are less costly but can violate user privacy. Heilym [19] used visual camera images to detect, extract human skeletal information, find the most salient skeletal features, and estimate human pose by using random forest (RF), support vector machine (SVM), four classifier models, multilayer perceptron (MLP), and k-nearest neighbor (KNN). Kong [20] used a depth camera to acquire a person’s standing or falling skeletal images and fast Fourier transform (FFT) to encrypt the images and perform fall detection. Depth-camera-based fall detection is superior compared to using RGB images in protecting users’ privacy, but this solution suffers from high equipment cost, narrow detection angle, and short effective distance.

#### 2.1.2. Ambient Sensors

Ambient-sensor-based fall detection methods are used to help identify fall behavior by monitoring changes in a specific region of interest (ROI) and define posture, movement, and human presence. These ambient sensors include infrared, acoustic, passive infrared, and Doppler radar. Hayashida [21] detected falls by emitting and capturing infrared radiation to measure the heat emitted by an object and obtain environmental changes in a certain ROI. Popescu [22] proposed an acoustic-based fall detection system that consists of a microphone array and motion detector that flags events as falls or ADLs using acoustic characteristics. Most falls have a significant vertical motion component. Liu [23] emitted microwave signals from multiple Doppler radars placed on the roof of a house. They can accurately measure the radial velocity of the target relative to the radar to determine the occurrence of a fall. While all these ambient sensors show good results, their use is limited to the ROI in which they are placed and their performance is affected by other users and environmental changes, such as heat or other devices.

### 2.2. Wearable Sensors

Unlike non-wearable sensors, wearable sensors hardly violate a user’s privacy and are less restricted by the area in which they are used. This type of sensor also has the advantages of low power consumption, high convenience, and low cost, and more fall detection research is based on wearable sensors. Such sensors are usually embedded in wristbands, belts, or other textile patches to monitor kinematic properties (acceleration, angular velocity, magnetic induction strength) obtained from accelerometers, gyroscopes, or magnetometers, etc., providing posture and motion information [7]. The type of wearable sensor, sampling rate, and placement are usually used as differentiating criteria. According to existing research, accelerometers and gyroscopes are the most frequently used wearable sensor types, with sampling at frequencies of 50 Hz, 100 Hz, and 200 Hz accounting for the majority. Most data collectors choose to place the sensor at the waist or wrist, with a few at the thigh or ankle [24]. Kusumah [25] implemented a threshold-based fall detection algorithm using accelerometer and gyroscope sensor signals. However, the actual fall process is more complex. Fall-like behavior (e.g., sitting and lying down) can cause significant interference to the algorithm. These sample values for fall-like activates can also reach the threshold limits set for fall settings at some point and thus have a high false alarm rate. Deep-learning-based classification recognition has currently been used in more fall detection algorithms. He [26] proposed a fall detection convolutional neural network (FD-CNN) model with higher detection specificity compared to the traditional threshold method. Three-axis acceleration was normalized and mapped to RGB bitmaps. Spatial features were extracted using FD-CNN networks to detect fall and no-fall behavior based on feature information in the bitmaps. Yhdego [27] transformed acceleration data into scalar maps and used time–frequency analysis and a pre-trained deep convolutional neural network (DCNN) model for transfer learning. The fall data are continuous and convolutional processing can result in the loss of time-dimensional features. In order to solve this problem, Mirto [28] used a long short-term memory (LSTM) fall detection algorithm that automatically extracted temporal information using long time sequences as input. The LSTM structures take into account the temporal and spatial relationships of the data and can accurately distinguish between fall-like and true fall behavior. This kind of method often requires a large amount of calculation and has a complex model structure, with a concomitant increase in time complexity, making detection inefficient. The fall detection method proposed by Yhdego [29] used an attention mechanism to observe sensor data peaks. Falls can have multiple peak data in a short period of time. Focusing only on local peaks is susceptible to interference from fall-like behavior, giving the model low specificity. Regarding wearable devices as the main tool of fall detection, there are some problems with the advantages of convenience and low cost. These sensors expect users to charge them regularly and carry them with them all the time. This does not always happen, however, because the main users of these sensors are elderly people. On the other hand, these wearable devices have limited computing power, and deploying neural network models on them requires consideration of limitations, such as device memory capacity, device power duration, and user wearing comfort. Also, the detection time of the model is required to reach a low value in order to meet the timeliness requirement of fall detection.

## 3. Materials and Methods

As is shown in Figure 1, the framework proposed in this study comprises two steps: data preprocessing and feature extraction and classification based on the Patch-Transformer Network (PTN) algorithm. Specifically, in the first step, the input wearable device data are first normalized in size to ensure the input format is the same. The sliding window is used to divide the data of different channels into subsequences, and then the subsequence data of different channels are concatenated. Each subsequence datum is reshaped to meet the input requirements of the PTN algorithm.

In the second step, the local feature information of the input sequence is extracted by CNN, and the global feature is extracted through the Transformer encoding layer. Global pooling enhances the correlation between features. The final classification result is derived from the linear layer.

### 3.1. Datasets

Public datasets SisFall [30] and UnMib SHAR [31] are used as our data sources to test the accuracy and latency of the model. These two datasets include different types of falls and ADLs.

The SisFall is a dataset that, on the posture of volunteers under different activities, was assembled by device consisting of accelerometer and gyroscope. The data include 19 ADLs and 15 types of falls performed by 23 young people, 15 ADLs performed by 14 healthy independent participants over the age of 62, and data on all ADLs and falls types performed by 1 participant aged 60.

UnMib SHAR is a fall and no-fall sample dataset collected by the accelerometer sensor in the Android smartphone. Limited by the Android operating system, UnMib SHAR used sampling frequency of 50 Hz. The dataset consisted of 11,771 samples of human ADLs and falls events. The participating volunteers were between the ages of 18 and 60 years and the samples were divided into 17 sub-categories and 2 coarse categories: one containing nine ADLs samples and the other including eight samples of falls.

### 3.2. Data Preprocessing

In the SisFall and UnMib SHAR datasets, the data sample size was not uniform, and regular activities, such as slow walking and jogging, were present in both datasets. These activities were sampled for longer periods of time compared to the other behavior. To obtain falls and ADLs datasets with uniform data specifications and to ensure data integrity, we process the data accordingly for both datasets. In SisFall, the entire ADLs are sampled with a 12 s duration. We split the longer time series activities into a few equal length subsegments, each of 12 s duration, whereas, in the fall samples, the sample sizes are all 15 s. The fall dataset is downsampled to reshape the sample size to 12 s. The sampling frequency of UnMib SHAR is different from SisFall. The segmented fragment size of the long time series samples is 3 s. The segmented ADLs subsequences, downsampled fall samples, and unprocessed raw data are recoded to obtain a falls and ADLs dataset with uniform data size. The SisFall dataset includes 3778 ADLs and 1798 falls samples, each with a sample size of 12 s. UnMib SHAR includes 4191 falls and 7578 data samples of 3 s duration. We use overlapping sliding time windows to segment the data, with each time window being *W* samples long, and reshape the samples into rectangular data blocks. The whole sequence is partitioned into a sequence of data blocks of length *N* by gradually moving backwards with a fixed sliding step *S*. If the sliding step is too large, the segmentation will result in the loss in correlation between time series, so the moving step of the sliding window needs to be smaller than the width of the time window. If the sliding step is too small, it will cause problems, such as serious data overlap and inconspicuous distinction of local information. Figure 2 illustrates the process of sliding window segmentation of subsequence of our multi-channel data from acceleration sensors.

### 3.3. Patch-Transformer Network Algorithm

We propose a fall detection method called PTN by combining the deep learning algorithms of CNN and Transformer Network. An overview of the method is depicted in Figure 3. In practical fall detection, multiple sensors are often used to determine posture information by placing them at different locations on the human body. In this paper, we describe the PTN model using three sensors as examples, where $S_{Device1},S_{Device2},S_{Device3}$ denote the sequence of data blocks of three different sensors after sliding window segmentation of the data and $P_{i}\in R^{n^{2}\times C}$ is a single data block for sensor. Each data block contains the x, y, z channel values of the sensor. Input sequence *I* is convolved by two convolution layers to extract local features for each matrix block (the data are extracted from each sliding window and reshaped into the data matrix of the n×n) sequence fall, with a convolution kernel of size 3 and a step size of 2, and the ReLU activation function is used. Then, local features are flattened and transposed to obtain the fall local feature mapping sequence matrix $F_{ND}\in R^{D\times N}$($D=C^{2}$, *C* is the number of channels after merging multiple sensors). The input sequence needs to add the position encoding information $E_{pos}$ of the sequence before being fed into the encoding layer to ensure that the time series information is not lost during the network computation. We only use the encoding layer part of the Transformer structure for the classification task of fall detection. The sequence data with the position encoding were added and fed into the Transformer encoding layer.
(1)$$\begin{array}{c} I=Concat(S_{Device1},S_{Device2},S_{Device3}) \end{array}$$
(2)$$\begin{array}{c} \left[Q,K,V]=X[W^{Q},W^{K},W^{V}\right] \end{array}$$
(3)$$\begin{array}{c} SA\left(X\right)=softmax\left(QK^{T}/\sqrt{d_{k}}\right)V \end{array}$$

In this model, the projections $W^{Q}\in R^{d_{model}\times d_{k}},W^{K}\in R^{d_{model}\times d_{k}},W^{V}\in R^{d_{model}\times d_{v}}$ are parameter matrices. The *h* is the number of heads of multi-head self-attention (MSA). MSA allows the model to learn feature information from the features extracted from the different convolution kernels. In this paper, the projections $W^{O}\in R^{h\times d_{model}\times d_{v}}$ are parameter matrices, $d_{k}=d_{v}=d_{model}/h$, and $d_{model}=D$, where *D* of Sisfall and UnMib SHAR dataset are 9×9 and 3×3, respectively.
(4)$$\begin{array}{c} MSA\left(X\right)=Concat\left(SA_{1}\left(X\right),SA_{2}\left(X\right),\cdots ,SA_{h}\left(X\right)\right)W^{O} \end{array}$$

The encoding layer consists of *L* sub-encoding layer. Each of them is followed by a residual connection structure [32], which avoids the problem of gradient information loss and accuracy degradation as the depth of the network increases during the cyclic computation of the sub-encoding layers. The sample data representation in sub-encoding layer, as the input to a multi-layer perceptron (MLP), Y uses the output of the entire Transformer encoding layer. Our encoding layer outputs global average pooling (GAP). The final fall and no-fall classification results are provided by a single linear layer.
(5)$$\begin{array}{c} X_{l}^{\prime }=MSA\left(LN\left(X_{l-1}\right)\right)+X_{l-1},\;l=1,2,\cdots ,L \end{array}$$
(6)$$\begin{array}{c} X_{L}=MLP\left(LN\left(X_{l}^{\prime }\right)\right)+X_{l}^{\prime },\;l=1,2,\cdots ,L \end{array}$$
(7)$$\begin{array}{c} Y=LN(X_{L}) \end{array}$$
(8)$$\begin{array}{c} Y^{\prime }=GAP\left(Y\right) \end{array}$$
(9)$$\begin{array}{c} Probability=Linearlayer(Y^{\prime }) \end{array}$$

## 4. Results

We performed experiments with different datasets; the training was performed on the SisFall dataset to verify the model validity. We used the UnMib SHAR fall dataset to verify the scalability of the model. Our proposed PTN model is experimentally compared with existing fall detection techniques that perform better. All the experiments in this paper are performed on a PC device with an Intel Core i5-12400 CPU and NVIDIA GeForce GTX 3080 GPU. The software environment is Python 3.8.13 with PyTorch 1.12 CUDA version 11.3.

### 4.1. Training and Testing Strategy

We choose the first 80% of the data for training the model and the remaining 20% of the set for model validation. The model is trained by a 16 and 32 mini-batch for 50 epochs. The best epoch with minimal loss was used as the results for evaluation. We use a learning rate decay strategy to prevent overfitting. Table 1 shows the training hyperparameter settings in detail.

### 4.2. Evaluation Indicators

Confusion matrices are commonly used to analyze classification and prediction models. The fall detection confusion matrix used in this paper is shown in Table 2. To evaluate the performance of the model, we introduce accuracy (Acc) as the metric to judge the detection efficiency of the model, sensitive (Sen) to prevent false positives, and specificity (Spe) to describe the capability of the model for detecting falls. We add F1−score into our training metric. Based on the confusion matrix, the metrics above can be expressed as follows (14)–(17).
(10)$$\begin{array}{c} Acc=\frac{TP+TN}{TP+TN+FP+FN}\times 100\% \end{array}$$
(11)$$\begin{array}{c} Sen=\frac{TP}{TP+FN}\times 100\% \end{array}$$
(12)$$\begin{array}{c} Spe=\frac{TN}{TN+FP}\times 100\% \end{array}$$
(13)$$\begin{array}{c} F1-score=\frac{2\times \frac{TP}{TP+FP}\times Sen}{\frac{TP}{TP+FP}+Sen}\times 100\% \end{array}$$

### 4.3. Model Parameters

There are usually parameters in neural networks that cannot be learned automatically by the model and need to be set manually before training. In this paper, we use Transformer encoding layers in which the number of parallel attention heads and the number of sub-encoding layers have not yet been determined. Based on previous work [33] and the wearable sensors data size, we find that, as the number of attention heads and sub-encoding layers increases, this does not obviously improve the model accuracy in experiments. Most of these parameters are redundant during the training model period, especially for small-input data dimensions. The increase in model parameters will cause longer training and testing time. Model size and computational complexity also expend, which is not beneficial for the fall detection task. To balance the model size and performance metrics, we employ a small Transformer encoding layer structure, setting the parallel attention heads *h* to 3 and the number of sub-encoding layers L to 6. The model structure with this combination of parameters is used in all subsequent evaluations of the results.

### 4.4. Ablation Experiments

Falls occur when people are performing an activity or are suddenly disturbed by the external environment and their duration is short. Using sliding window to intercept sample data can fit the local feature information at the moment of the fall. Different window sizes have a direct impact on model accuracy. To determine the optimal value of the sliding window, we conducted sliding window ablation experiments on both SisFall and UnMib SHAR datasets. As the sampling frequencies and sample sizes of the two datasets were different, we chose different window sizes for the control group. In the SisFall dataset, we set sliding windows for 1 s, 2 s, 4 s, and 6 s, and correspondingly used window sizes for 1 s, 2 s, and 3 s in the UnMib SHAR dataset. Table 3 details the performance of the model with different window sizes for both datasets. The performance of metrics Acc, Sen, and F1-score all perform best with a time window of 2 s on the SisFall dataset, while the specificity Spe is slightly lower than the value for the 4 s window; on the UnMib SHAR dataset, again, the sliding window size of 2 s works best, with Spe slightly lower than the performance for the 1 s window. In the seven and eight columns of Table 3, we also list the number of model parameters and the number of operations. The 2 s window size still has a lower computational complexity and parameters computed while achieving the best performance. In fact, the fall occurrence duration is close to 2 s [26], and an accurate fit of the feature information of fall occurrence can improve the model detection accuracy. Therefore, 2 s sliding time window has a strong feature expression ability, and the performance of metrics parameters is outstanding. On the other hand, according to the last column of the table, a PTN algorithm running on a PC with GTX3080 graphics card only spends about 4 ms on the SisFall datasets and spends about 1 ms on the UnMib SHAR dataset to classify, which can fully fit the need of real-time fall detection.

### 4.5. Comparison with Existing Methods

In this section, we compare the performance with existing fall detection methods, and the experimental results are completed on the SisFall and UnMib SHAR datasets. We compare the experimental results of others based on accelerometer and gyroscope fall detection methods FD-CNN, CNN+XGB, BiLSTM, and SVM. We also list fall detection methods that employ one signal type, CNN+LSTM and TBM+CNN. As shown in Table 4, on the SisFall dataset, we achieved event-based Acc, Sen, and Spe of 99.86%, 100.0%, and 99.8%, respectively, as well as a higher F1-score of 99.45%. Our method has 2.45% higher Acc and 4.31% higher Spe compared to BiLSTM, which is much higher than the performance metrics of the fall detection method proposed by CNN-XGB; on the UnMib SHAR dataset, our method also achieves an Acc of 99.14%, Sen 98.85%, Spe 99.31%, and F1-score of 98.83% for the experimental results. All the parameters are higher than in the TBM+CNN method. According to Table 4, the PTN fall detection method based on accelerometer gyroscope data proposed in this paper achieves the highest level of performance in all parameter metrics.

## 5. Discussion

### 5.1. Impact of Local Feature Extraction on Model

This paper introduces a neural network fall detection method based on the Transformer structure. The multi-head self-attention in the Transformer encoding layer is global, and each word can learn the feature information of the whole sequence. However, this structure itself lacks local and translation compared to CNN, and it has insufficient inductive understanding of local feature information. Wearable sensors fall data are a time series signal. The moment of fall occurrence and the whole process of the event exhibit both local and global feature information. Both features have a direct impact on the final detective results. We therefore use CNN to extract local feature information from the fall sample data, and automatically learn the global features of the sample using the Transformer multi-headed self-attention mechanism. Table 5 shows the ablation experiments for local feature extraction and analyzes the impact on the model. The results show that focusing on local features can improve detection accuracy, which also indicates that combining the inductive biasing capability of the convolutional structure with the global learning features of the multi-head self-attention mechanism can enhance the model’s ability to perceive the data and improve detection performance.

### 5.2. Model Complexity Analysis

Model complexity has a high reference significance for the task of fall detection. The higher the model complexity, the higher the computational power requirements of the hardware device. The performance and computing power of ordinary wearable devices are limited and cannot support large-scale neural network models. In addition, the detection time of large models is long and cannot meet the timeliness requirements of fall detection. Therefore, it is necessary to take model complexity into account in the method design. In the PTN model, we use the stack of small convolution kernels in two convolution layers instead of large convolution kernels, achieving the same feature extraction effect and obviously decreaseing the model parameters. The traditional Transformer encoding layer is a sentence, and each word in the sentence is mapped to a word vector. We choose a small feature projection vector dimension, and global average pooling (GAP) was used to receive the output of the encoding layer. The method has no learnable parameters, which not only reduces the amount of computation but also avoids overfitting. On these bases, the model complexity is reduced effectively, and the expected drop detection effect is achieved.

### 5.3. The Impact of Wearable Sensor Use on Algorithms

#### 5.3.1. Effect of Sample Distribution

Uneven distribution of training set and test set samples can lead to overfitting or underfitting problems. In the training data, some activities have a higher proportion of samples than others, while, in the test set, these activities have the same proportion; this situation will cause the model to be overfitted, making the test poor. Also, underfitting can occur in cases where the training and test samples are significantly different.

#### 5.3.2. Large-Scale Datasets Required

A Transformer model has much fewer inductive bias capabilities than CNNs. These inductive bias capabilities can provide prior knowledge of data features and improve model generalization. A Transformer model requires large-scale data for training to make up for these deficiencies. We use this inductive bias of convolution to help extract local features from the data, on which the Transformer model is trained, effectively alleviating the reliance of model on large-scale data. However, models that want to distinguish in detail between specific types of a certain activity still require a large number of datasets for training.

#### 5.3.3. Impact of Sensor Accuracy

Low-precision wearable sensors may be less able to distinguish between similar activities, reducing model specificity. For example, in the process of human squatting and lying down, the value of acceleration will rapidly increase and then level off. The amplitude of the action of these two activities is similar, and it is difficult to distinguish the specific type of activity of this sample in the measurement results of low-precision sensors. Therefore, this should be avoided as much as possible by using high-precision sensors.

## 6. Conclusions

This paper proposed a new fall detection method called Patch-Transformer Network, which uses convolution to extract local feature information in a sliding window segmentation data matrix and uses the Transformer multi-head self-attention mechanism to learn global features of fall behavior. The final detection result is output from a linear layer. The network model has good performance for time series in general and wearable sensor data. This work is better than previous attempts based on wearable sensors signals in various evaluation metrics. In the future, we will develop a fall detection system based on wearable sensors and deploy this model on the fall detection system to automatically detect the occurrence of fall behavior and reduce the workload of medical staff.

## Figures and Tables

**Figure 1 sensors-23-06360-f001:**
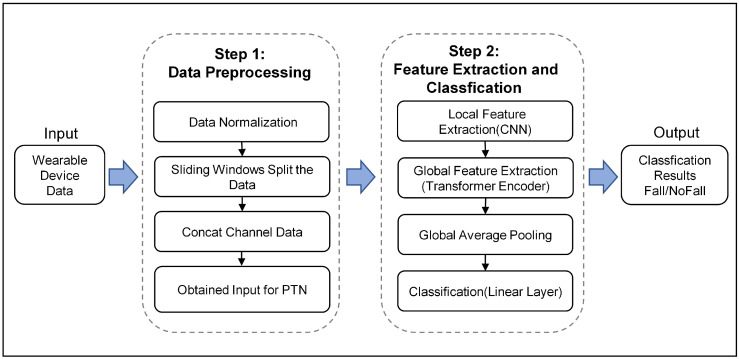
Block diagram of the workflow of the proposed method.

**Figure 2 sensors-23-06360-f002:**
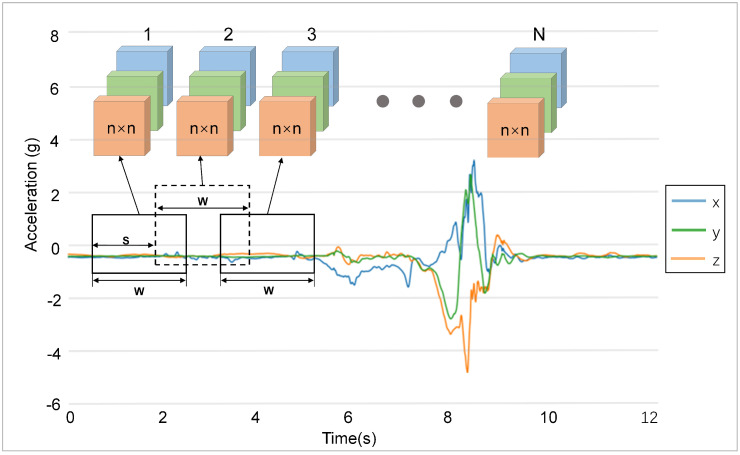
Sliding-window-segmentation-data-based acceleration sensors.

**Figure 3 sensors-23-06360-f003:**
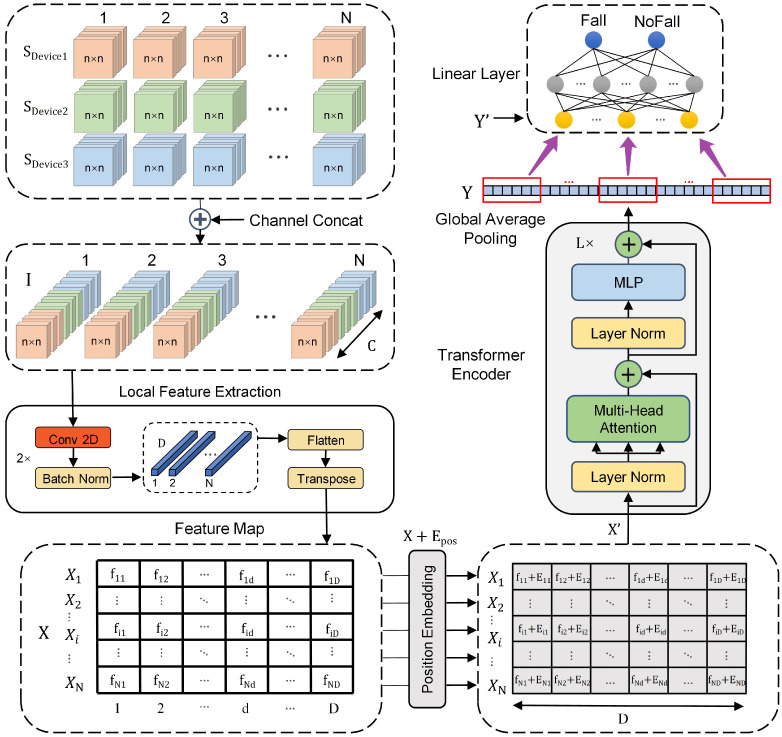
Patch-Transformer Network model overview.

**Table 1 sensors-23-06360-t001:** Experimental hyperparameter settings.

Parameters	SisFall	UnMib SHAR
Batch size	16	32
Epochs	50	50
Optimizer	Adam	Adam
Learning rate	0.001	0.001
Lr* decay	0.98	0.95

Lr* represents learning rate.

**Table 2 sensors-23-06360-t002:** Confusion matrix.

	Predicted Fall	Predicted No-Fall
Actual Fall	TP	FN
Actual No-Fall	FP	TN

FP, TP, FN, and TN denote false positive, true positive, false negative, and true negative, respectively. In the fall detection classification task, TP represents the samples in which a fall occurred and was correctly judged; TN represents ADLs that were correctly identified samples; FP represents the samples in which ADLs were judged as fall behavior; FN represents the samples in which falls were identified as ADLs.

**Table 3 sensors-23-06360-t003:** Model performance at different window sizes.

Datasets	Window Size	Acc (%)	Sen (%)	Spe (%)	F1-Score (%)	Params (M)	Flops (M)	TestTime (ms)
SisFall	1 s	97.90	96.84	98.85	97.71	0.64	5.77	4.51
	2 s	99.86	100.0	99.76	99.45	0.77	6.91	4.58
	4 s	99.50	98.94	99.80	99.43	1.05	9.43	4.80
	6 s	99.48	98.86	98.98	99.28	1.32	11.87	4.92
UnMib SHAR	1 s	98.97	99.30	99.31	98.47	0.06	0.50	1.36
	2 s	99.14	98.85	99.29	98.83	0.06	0.52	1.37
	3 s	98.42	97.02	99.13	97.64	0.07	0.56	1.43

**Table 4 sensors-23-06360-t004:** Performance comparison with existing fall detection methods.

Method	Dataset	Signal	Results
FD-CNN [26]	SisFall	Accelerometer Gyroscope	Acc = 98.61%
			Sen = 98.62%
			Spe = 99.80%
CNN+XGB [34]	SisFall	Accelerometer Gyroscope	Precision = 90.59%
			Sen = 88.25%
			Spe = 99.36%
			F1-score = 89.11%
BiLSTM [35]	SisFall	Accelerometer Gyroscope	Acc = 97.41%
			Sen = 100%
			Spe = 95.45%
			Precision = 94.28%
SVM [36]	SisFall	Accelerometer Gyroscope	Acc = 96.0%
			Sen = 99.0%
			Spe = 94.0%
CNN+LSTM [37]	UnMib SHAR	Accelerometer	Acc = 99.11%
TBM+CNN [38]	UnMib SHAR	Accelerometer	Acc = 97.02%
			Sen = 97.83%
			Spe = 96.64%
PTN (Ours)	SisFall	Accelerometer Gyroscope	Acc = 99.86%
			Sen = 100.0%
			Spe = 99.76%
			F1-score = 99.45%
PTN (Ours)	UnMib SHAR	Accelerometer	Acc = 99.14%
			Sen = 98.85%
			Spe = 99.29%
			F1-score = 98.83%

**Table 5 sensors-23-06360-t005:** Local feature ablation experiments.

Datasets	Local Feature	Acc (%)	Sen (%)	Spe (%)	F1-Score (%)
SisFall	-	97.01	96.01	97.85	96.70
	✔	99.86	100.0	99.76	99.45
UnMib SHAR	-	95.24	95.97	94.89	93.16
	✔	99.14	98.85	99.29	98.83

✔ represents that the PTN model contains local features.

## Data Availability

The fall repositories used in this study can be acquired online at https://github.com/BIng2325/SisFall/releases (SisFall) (accessed on 10 July 2023) and http://www.sal.disco.unimib.it/technologies/unimib-shar/ (UnMib SHAR) (accessed on 10 July 2023).

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
