# Peer review of "Patch-Transformer Network: A Wearable-Sensor-Based Fall Detection Method"

_sensors, 2023, doi:10.3390/s23146360_

Round 1

Reviewer 1 Report

This article presents a new algorithm for wearable sensor-based fall detection, named Patch-Transformer Network (PTN). The algorithm applies convolutional neural networks (CNN) to extract local features from fall data, and uses a Transformer encoding layer to learn global features from fall data. Then, it uses a linear layer to output the classification results of fall and non-fall events. The algorithm was evaluated on two public fall datasets, SisFall and UnMib SHAR, and achieved accuracy rates of 99.86% and 99.14%, respectively, outperforming existing fall detection methods. The algorithm also has low model complexity and computational overhead, making it suitable for deployment on wearable devices.

1. The training data consists of only 11771 samples of human ADLs and falls events. Is this enough for the transformer model?

2. How much time does the detection take, and does it meet the real-time requirements?

3. The method is validated by the existing data, but how does it perform on the real data from a wearable device?

4. Considering that a wearable device has low computing ability, can the proposed method be deployed on the wearable device?

Moderate editing of English language required.

Reviewer 2 Report

This manuscript proposed a deep learning-based method to identify body falling via a set of wearable sensors.  The algorithm is well presented. However, there's a key issue that the difficulties induced by wearable sensors are not presented quantitatively.  I suggest a minor revision to address this concern.  

In my previous report I raised a critical concern about the effect of wearable sensors compared to the traditional ones. Below are the detailed descriptions of this concern.

1)     The difference between wearable sensors and traditional sensors need to be specified quantatively. 

2)     What difficulties do wearable sensors bring to the algorithm? For example, low accuracy may knock-down the performance, etc.

3)     Any figures or videos to show the fall process and to convince the applicability of the proposed method?

Reviewer 3 Report

A new patch transformer network wearable sensor-based fall detection algorithm is proposed. The performance of the new method is compared to that of other methods. The system is very effective. The paper is well written and I think that it is suitable for publication in Sensors.

I made some corrections directly in the pdf version of the submitted ms that I am attaching to this email

I tried to revise the english but I would suggest a revision of a native english speaker scientist
